# LangDriveEdit: Language-Driven Image Editing for Street Scenes

## Abstract

Ensuring the safety of autonomous driving systems requires rigorous evaluation across diverse street scene conditions within the Operational Design Domain (ODD), such as lighting, weather, traffic, and road variations. Yet collecting real-world data to cover this spectrum is costly, time-consuming, and often impractical. Recent advances in language-driven image editing offer a promising alternative by simulating diverse scenarios through text-based modifications. However, progress has been limited by the absence of a dedicated dataset for driving-scene editing. To address this gap, we introduce, to the best of our knowledge, the first dataset specifically designed for **language-driven editing of driving scenes**. Our dataset combines real-world and synthetic street scene images and supports 12 distinct editing tasks, spanning global modifications (e.g., weather, season, time of day) and fine-grained local edits (e.g., altering vehicle or pedestrian attributes). Crucially, each edit is paired with **detailed textual and visual instructions**, and, together with our proposed supervised and unsupervised fine-tuning objectives, enables state-of-the-art image editing models to follow instructions faithfully and preserve critical content. Experimental results demonstrate that training language-driven editing models with our dataset and objectives yields substantial gains in prompt alignment, visual fidelity, generation realism, and downstream driving-task performance on edited street scene images, across diverse driving domains.

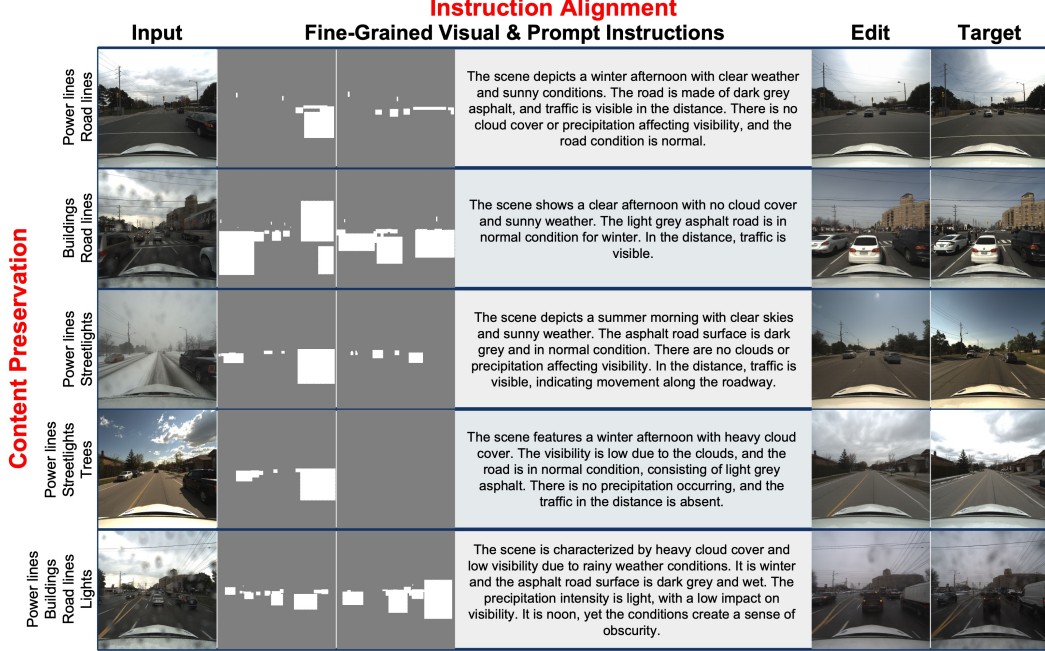

Figure 1: LangDriveEdit targets improving two critical requirements in driving scene editing: **content preservation** and **instruction alignment**. These two requirements are supported via fine-grained visual and prompt instructions (Sec. 3) and carefully designed training objectives (Sec. 4). Given a mask of dynamic objects to remove and add respectively, and a global editing prompt, we visualize our edits of given input images.

# 1 INTRODUCTION

Ensuring safety is a central challenge in deploying autonomous driving. Real-world testing within the Operational Design Domain (ODD) is limited by dynamic factors such as lighting, weather, traffic, and road conditions, making comprehensive data collection infeasible (Mehlhorn et al., 2023). Generative models offer a scalable alternative for synthesizing diverse environments (Gao et al., 2023), and instruction-guided image editing in particular enables fine-grained, language-based control while preserving realism through large-scale pretrained sources (Ramesh et al., 2022; Betker et al.; Rombach et al., 2022; Brooks et al., 2023b; Zhao et al., 2024).

Nonetheless, two properties critical to autonomous driving scenes are not explicitly enforced when applying generative instruction-guided editing: **Content Preservation** and **Instruction Alignment**. **Content preservation** focuses on retaining the unedited elements of a driving scene during transformation (Shi et al., 2024). Editing models may unintentionally modify or remove essential content—such as traffic signs, lane markings, or surrounding vehicles, potentially leading to incorrect visual signals for downstream perception and planning systems by altering safety-critical elements. Meanwhile, **instruction alignment** refers to the accurate execution of detailed, multi-attribute natural-language instructions (Shi et al., 2024). Strong instruction alignment not only supports human-in-the-loop workflows but also enables systematic exploration of the combinatorial space of scene factors (weather, illumination, traffic composition, viewpoint, and beyond), ensuring that each syntactic variation of an instruction is faithfully realized in the output, thereby achieving a degree of diversity and granularity unattainable through unguided random sampling.

Therefore, we pose two fundamental questions for current instruction-guided editing models:

> **Q1**: How can instruction-guided editing models generate variability in the environment while preserving unedited portions of a driving scene unchanged?
> **Q2**: How can instruction-guided editing models' generation be controlled given precise editing instructions in driving scenes?

The primary bottleneck in addressing these questions lies in the lack of **paired datasets** of **fine-grained visual or textual prompts**. We introduce the *LangDriveEdit* Dataset (Figure 1). It includes large-scale paired real-world driving images with fine-grained instructions (Sec.3.1), with a supplementary synthetic part (Sec.C.1). Our **real-world** images capture **a large amount of environmental variations**, such as season, lighting, and weather, alongside multiple concurrent object-level differences among road users (Sec.3.1). To enable fine-grained control in diffusion-based generation, we pair precise editing instructions with pixel-level masks, created using large language models integrated into a multi-modal vision pipeline. Unlike object-centric image generation, traffic scenes are densely populated with vehicles, pedestrians, and buildings, making natural language prompts alone insufficient (Figure 3). Our masks encode localized semantics at the pixel level, ensuring accurate description and manipulation of complex driving scenes. Furthermore, to scale driving scene editing to various traffic conditions, we introduce unsupervised training methods that encourage content preservation via cycle and identity objectives, and also instruction alignment via CLIP similarity objectives and adversarial training. We demonstrate that models trained this way on the *LangDriveEdit* dataset achieve strong **content preservation** and **instruction alignment** (Sec.5). Our contributions can be summarized as follows:

1. **New Paired Datasets for Driving Scene Editing.** To the best of our knowledge, *LangDriveEdit* is the first dataset of large-scale paired images with the support of diverse editing types and fine-grained instructions, designed for instruction-guided editing of driving scenes.
2. **Automatic Generation of Prompts and Visual Masks as Instructions.** Our pipeline streamlines the generation of editing prompts and pixel-level masks for *both* real-world and synthetic environments, employing a novel annotation framework that leverages vision-language models and depth estimation to extract environmental and object-level details.
3. **Significant Improvements in Driving Scene Editing.** Across different state-of-the-art editing models, our comprehensive experiments demonstrate that after our fine-tuning on our *LangDriveEdit* dataset, both **content preservation** and **instruction alignment** are largely improved.
4. **Impact on Downstream Driving Tasks.** We demonstrate that edited images produced with our dataset can **improve road segmentation performance** on an out-of-distribution driving dataset, highlighting the potential of instruction-guided editing for safety-critical applications.

## 2 RELATED WORK

**Image Editing Dataset.** Building image-editing datasets is more challenging than Text-2-Image (Betker et al.; Ramesh et al., 2021; 2022; Rombach et al., 2022), with data scarcity a key bottleneck (Wang et al., 2023a; Hui et al., 2024). Existing efforts such as MagicBrush (Zhang et al., 2024b), InstructPix2Pix (Brooks et al., 2023a), and SeedEdit (Shi et al., 2024) either rely on manual annotation, synthetic data, or iterative refinement, but remain limited to simple object edits. **Image Editing via Generation.** Advances in large diffusion models (Kawar et al., 2022; Saharia et al.; Chen et al., 2023) have enabled instruction-driven editing, with methods like InstructPix2Pix, HIVE, and UltraEdit pushing the field forward, though primarily in generic domains. **Image Editing for Autonomous Driving.** Driving-scene editing has been explored through NeRFs, Gaussian Splatting, and multi-condition generation (Liang et al., 2025; Gao et al., 2023), yet instruction-guided editing remains underexplored due to the absence of datasets. We address this gap with *LangDriveEdit*, the first instruction-driven editing dataset tailored to autonomous driving. Due to space limits, we refer readers to Section A for a more comprehensive survey of related works.

## 3 LANGDRIVEEDIT DATASET

The construction of the dataset involves a detailed annotation process to capture varying edits in driving scenes while maintaining consistency. We overview our dataset construction pipeline in Figure 2. In this section, we explain our data collection and annotation.

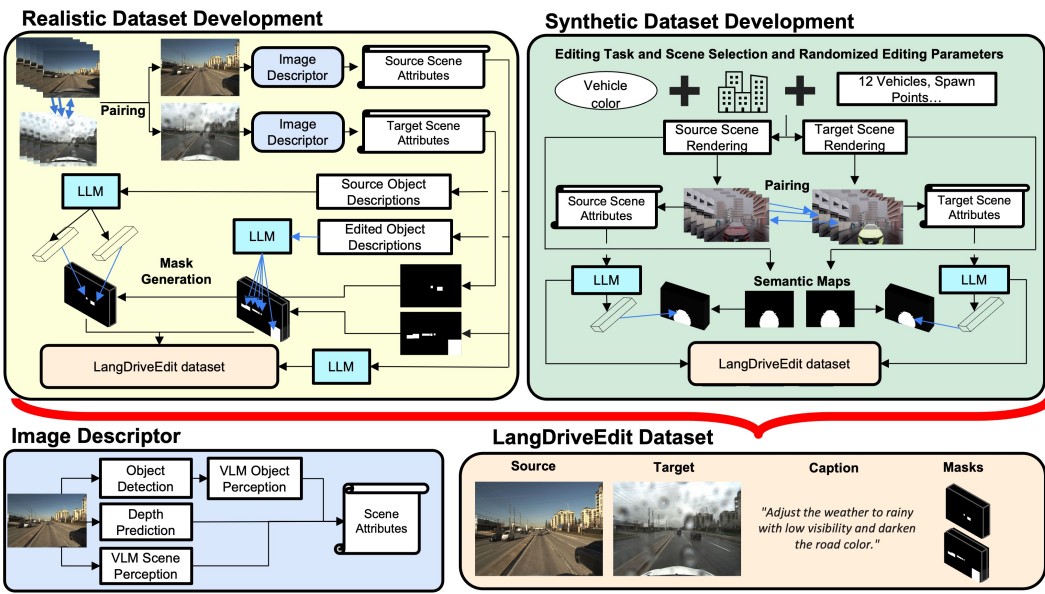

Figure 2: **LangDriveEdit Construction**. Real-world data are paired by camera pose and annotated using an image descriptor pipeline (Sec. 3.1.1) passed to an LLM to produce instructions (Sec. 3.1.2). For synthetic image pairs we simulate two frame sequences performing one of 12 editing tasks (Sec. C.1.1). Our dataset is composed of image pairs, editing instructions, and two masks indicating objects to add and to remove.

### 3.1 SEMANTIC ALIGNMENT FOR UNPAIRED REAL-WORLD DRIVING ENVIRONMENTS

Our primary data contribution consists of large-scale paired real-world driving images with hierarchical, fine-grained annotations. As demonstrated in the experiments (Section 5), these high-quality driving scene images form the foundation for training image editing models to generate diverse driving environments. We show our dataset statistics in Table 1.

We align images and semantics in Boreas (Burnett et al., 2023), which is a multi-season autonomous driving dataset collected by driving a repeated route throughout one year. In addition to ideal climates, Boreas features adverse weather conditions (rain, snow, fog) that are critical for rigorously evaluating and expanding operational design domains.

Table 1: **Real-World Dataset Statistics**. Left: distribution of edit types across the Boreas. Right: distribution of the number of edit types per example.

| Edit type | Count | Percentage | Partition by Number of Subedits | | |
|---|---|---|---|---|---|
| Road Conditions | 146,159 | 15% | 1 Edit Type | 16,623 | 4.86% |
| Time of Day | 235,710 | 24% | 2 Edit Types | 95,295 | 27.88% |
| Traffic | 263,064 | 27% | 3 Edit Types | 160,413 | 46.93% |
| Traffic Light | 55,998 | 6% | 4 Edit Types | 64,385 | 18.84% |
| Weather | 269,715 | 28% | 5 Edit Types | 4,922 | 1.44% |
| | | | **Total Examples** | 341,796 | - |

### 3.1.1 FINE-GRAINED PARING AND DECOMPOSITION OF COMPLEX DRIVING SCENES

Real-world images in Boreas are largely collected from unpaired scenes and camera poses. To align images into paired scenes with aligned camera poses, for each pair of driving sequences, we extract corresponding frames by minimizing camera pose disparities, below a fixed tolerance, employing an equally weighted sum of angular orientation and Euclidean position. We show this in Equation 1 where $x$ is the camera's Euclidean position and $\phi, \theta, \psi$ are its roll, pitch, and yaw respectively.

$$I_{target} = \arg \min_{I \in \mathcal{I}} \text{dist}(I, I_{source})$$
$$\text{dist}(I_a, I_b) = \|\vec{x}_a - \vec{x}_b\|_2 + |\phi_a - \phi_b| + |\theta_a - \theta_b| + |\psi_a - \psi_b| \tag{1}$$

where $\mathcal{I}$ denotes the neighboring frames of $I_{source}$, and $a, b$ are the indices of two such frames. To collect hierarchical and multimodal scene descriptions, we introduce "Image Descriptor" (Figure 2), a training-free pipeline inspired by (Yao et al., 2025). Real-world driving scenes introduce significant annotation challenges. *First*, most large-scale real-world recordings lack multimodal sensors, leaving RGB-based global and instance-level descriptions less informative. *Second*, the real world typically includes complex scene variations and compositions (see Figure 3 left, where images are captured with the same camera pose at different times).

Our "Image Descriptor" is a **comprehensive annotation system**: it integrates vision-language and depth estimation models that can generate semantic-aligned scene descriptions at two levels:

**Multimodal Environments Descriptions.** We first extract global information about the scene:

1. We use an image-based vision-language model (VLM) (Chen et al., 2024a) for a global interpretation of extremely fine-grained attributes. We show the VLM prompt in Appendix E.1.

2. To estimate object distances, we apply a metric depth estimation model, Metric3d (Hu et al., 2024), to the full image, producing a depth map whose values correspond to real-world distances.

**Instance-Level Semantic Decomposition.** After preparing the global description, we then record objects present in the scene:

1. We run a 2D object detector (Owlv2 (Minderer et al., 2024)) that returns, for each detected object, a bounding box, a class label (from the set 'ambulance', 'bicycle', 'traffic light', 'traffic cone', 'person', 'car', 'motorcycle', 'bus', 'building', 'fire truck'), and a unique object ID.

2. For each object, we crop the global depth map (from the 2nd step above) to its bounding box and then refine that region with a binary mask from the Segment Anything Model (SAM (Kirillov et al., 2023)), ensuring we exclude background pixels. The object's distance is taken as the mean depth over this masked area.

3. We invoke the VLM (Chen et al., 2024a) on each object's bounding box to extract additional attributes, such as vehicle color or traffic-light state.

We show an example annotation in Appendix E.

### 3.1.2 DENSE EDITING INSTRUCTIONS AND CONTENT PRESERVATION

We enable controllable image editing, with uninstructed content preserved, by using two annotation-driven guidances: textual instructions and spatial masks.

**Instruction Generation.** Using the global scene annotations we prepared in Section 3.1.1, such as weather, time of day, and road conditions, we employ ChatGPT 4o-mini to generate structured editing instructions separately for the input source image and the target driving scene (App. F.2). Our generated instruction only describes what the target image should look like.

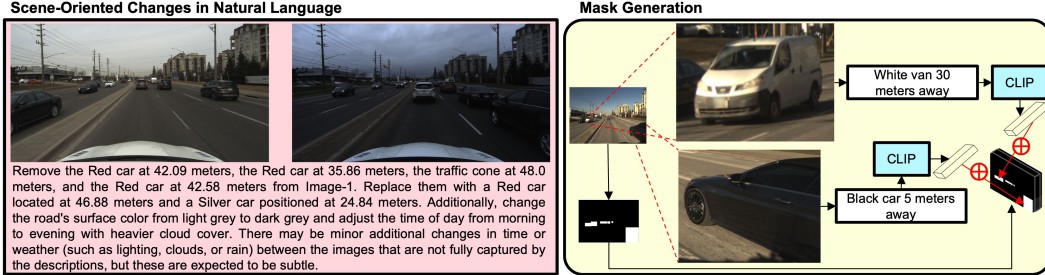

Figure 3: **Driving Scene Features are Dense**. Left: It is difficult to describe rich semantics for multiple object-oriented changes in natural language. Right: To provide fine-grained instructions with rich semantics, we construct both binary removal/addition masks and CLIP text features of instances (Section 3.1.2).

**Extracting Semantic-Rich Removal-Addition Masks.** To capture nuanced scene edits, each global instruction is paired with two instance-aware masks. Specifically, we generate: (1) a *removal mask* for the source image, which identifies objects to be eliminated, (2) an *addition mask* for the target image, which highlights regions designated for new object placement. Together, they support fine-grained object edits while preserving non-targeted regions. See details in Appendix D.

**Expanding Masks with CLIP Text Features.** Unlike traditional image editing approaches that primarily target sparse, object-centric modifications (e.g. (Zhao et al., 2024; Brooks et al., 2023a)) driving scenes are inherently much denser and more complex. It is unrealistic to precisely describe rich semantics for multiple instance-specific changes in natural language prompts. To enable the accurate execution of detailed multi-attribute instructions, we use masks expanded with CLIP features (Radford et al., 2021). For each masked object, we encode its text description into CLIP and assign the encoded feature to each masked pixel. This process is shown in Figure 3 right. Objects are processed in descending order of their distances to the ego camera, allowing the CLIP features of closer objects to overwrite those of farther, occluded objects at overlapping pixels.

## 4 LANGUAGE-GUIDED IMAGE EDITING

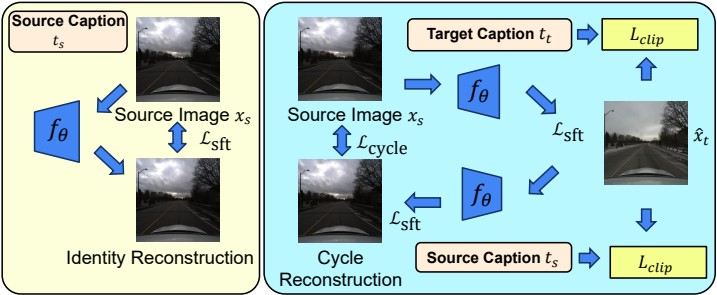

Figure 4: Training language-guided driving scene image editing. Our training pipeline supports both supervised training for paired images and unsupervised training for unpaired ones (e.g. downstream unseen real scenarios). We include three training objectives: supervised fine-tuning $\mathcal{L}_{\text{sft}}$ (Section 4.1), cycle consistency $\mathcal{L}_{\text{cycle}}$ (Section 4.2), and $\mathcal{L}_{\text{clip}}$ (Section 4.3).

In this section, we introduce a suite of training objectives to explicitly encourage **content preservation** and **instruction alignment**. Our training pipeline integrates both supervised and unsupervised objectives, enabling editing models to benefit from paired data for precise editing control when available, while remaining applicable to large-scale, unpaired datasets. In Section 5, we show that this supports the fine-tuning of different image editing models.

### 4.1 SUPERVISED FINE-TUNING FOR INSTRUCTION ALIGNMENT

When paired source–target examples are available, we train image editing models with supervised fine-tuning. Given paired training samples $(x_s, x_t, t_s, t_t, M_r, M_a)$, the generator model $f_\theta$ produces an edited image, and we calculate the supervised fine-tuning loss in Eqn. 2.

$$\hat{x}_t = f_\theta(x_s, t_t, M_r, M_a),$$
$$\mathcal{L}_{\text{sft}} = \lambda_{\text{sft}} \left\| x_t - \hat{x}_t \right\|_1 + \lambda_{\text{sft-lpips}} \left\| \phi(x_t) - \phi(\hat{x}_t) \right\|_2, \tag{2}$$

Each training instance includes a source scene ($x_s$), two natural language instructions describing the transformations from the source to the target ($t_t$), from target to the source ($t_s$), a mask of instances to remove and add from the source scene ($M_r, M_a$), and the resulting edited target ($x_t$). $\hat{x}_t = f_\theta(x_s, t_t, M_r, M_a)$ is the generator's output. $\phi(\cdot)$ denotes the feature extraction function of a pretrained VGG network (Zhang et al., 2018).

With this design, we can explicitly guide the model toward faithful instruction following. By conditioning the model on instruction-derived pixel masks, we constrain modifications to specified areas, encouraging the model to localize edits. This ensures unedited structures, such as road geometry and lane markings, are unchanged while surrounding vehicles and global features are edited. Masked supervision penalizes deviations in non-edited regions, which also supports content preservation.

As a special case of $\mathcal{L}_{\text{sft}}$, when $x_s$ and $x_t$ are the same image (with blank removal/addition masks), we are essentially asking the editing model to preserve the content:

$$\mathcal{L}_{\text{sft}} = \lambda_{\text{id}} \left\| f_\theta(x_s, t_s, \emptyset, \emptyset) - x_s \right\|_1 + \lambda_{\text{id-lpips}} \left\| \phi(f_\theta(x_s, t_s, \emptyset, \emptyset)) - \phi(x_s) \right\|_2 \quad \text{(identity preservation)} \quad (3)$$

This special case of $\mathcal{L}_{\text{sft}}$, i.e, an identity objective, enforces that when editing instructions correspond to no change (e.g., blank masks or re-adding removed content), the model reproduces the input. This teaches the model to preserve dynamic scene instances that are not specified in the editing instruction.

## 4.2 LANGUAGE-GUIDED CYCLE CONSISTENCY AND IDENTITY PRESERVATION

While supervised fine-tuning provides precise control, acquiring paired data is costly, and the reliance on paired data limits its scalability to driving datasets where explicit ground truth edits are unavailable, especially on unseen driving scenes in the wild. Therefore, we choose to include complementary unsupervised constraints via cycle consistency and identity preservation, such that we can use them in OOD unsupervised cases.

Cycle consistency extends this principle by encouraging reversibility. Without additional constraints, generative editing models may alter portions of the scene in regions unrelated to the instruction. By requiring the original image to be recoverable after a forward–backward editing cycle, the model is penalized for unnecessary deviations from the input. This encourages content preservation by discouraging drift.

$$\hat{x}_s = f_\theta(f_\theta(x_s, t_t, M_r, M_a), t_s, M_a, M_r)$$
$$\mathcal{L}_{\text{cycle}} = \lambda_{\text{cycle}} \left\| \hat{x}_s - x_s \right\|_1 + \lambda_{\text{cycle-lpips}} \left\| \phi(\hat{x}_s) - \phi(x_s) \right\|_2. \quad (4)$$

By combining pixel-level L1 and perceptual LPIPS losses, we enforce structural fidelity while allowing stylistic variation.

## 4.3 LANGUAGE-GUIDED CLIP LOSS FOR CONTENT PRESERVATION

Reconstruction-based supervision is insufficient on its own: the model may collapse to an identity mapping, avoiding all edits to minimize loss. To overcome this degeneracy, we incorporate a complementary alignment signal based on language–image similarity, described in the following subsection.

$$\mathcal{L}_{\text{clip}} = \lambda_{\text{clip}} \left(1 - \text{sim}_{\cos}\left(\text{CLIP}_I(\hat{x}_b), \text{CLIP}_T(t_b)\right)\right) + \lambda_{\text{clip}} \, \text{sim}_{\cos}\left(\text{CLIP}_I(\hat{x}_b), \text{CLIP}_T(t_a)\right), \quad (5)$$

where $\text{CLIP}_I$ indicates CLIP's image feature, $\text{CLIP}_T$ is for CLIP's text feature, and $\text{sim}_{\cos}$ for cosine similarity. To counteract degraded generation, in Equation 5, we first incorporate a language-guided CLIP similarity loss (the first term on right-hand side). This loss measures the alignment between the generated output and the provided instruction using CLIP. The aligned CLIP loss encourages outputs to move toward the intended semantic edit, thereby reinforcing instruction alignment. We also introduce a misalignment penalty term (the second term on right-hand side), which discourages similarity to input image description to prevent the model from reproducing the input.

## 5 EXPERIMENTS

### 5.1 SETTINGS

We evaluate our training strategies and pixel-level instructions on two competitive image editing models: UltraEdit (Zhao et al., 2024) and CycleGAN-Turbo (Parmar et al., 2024). Both models

are fine-tuned from powerful diffusion backbones trained on large-scale datasets. Following the evaluation protocol in (Zhang et al., 2024c), we assess editing performance using L1 distance, L2 distance, CLIP image similarity, and DINO similarity. Additional implementation details are provided in App. G.

We structure our experiments to address two core questions in Section 1: 1) The importance of paired driving scene data and training objectives for generating desired edits while maintaining scene integrity (Sec. 5.2). 2) The extent to which fine-grained prompting enhances models' generation to align with instructions and handle the complex interplay of foreground and background modifications in driving scenes (Sec. 5.3). In Sec. 5.4 we extend our editing to out of distribution images and evaluate the downstream performance of a road segmentation model using our edits.

## 5.2 Controlling Generation via Precise Editing Instructions

We first study the importance of paired driving scene images and training objectives for performing driving scene editing. Our quantitative results in Table 2 show that models trained with our precise editing instructions consistently outperform their baselines. Our trained models ("Ours") trained with Removal-Addition masks (Sec. 3) and training objectives (Sec. 4) show the lowest L1 and L2 scores which suggest high content preservation in unedited regions, and high instruction following in edited regions. Furthermore their high CLIP and DINO scores indicate the strongest instruction alignment while preserving the scene content. Qualitatively, Figure 5 shows that the base CycleGAN-Turbo often fails to modify the images following the pixel guidance (rows 1, 2, 6). Bagel, while it does not support masks, fails to preserve the scene content (rows 3, 4, 6), and UltraEdit, while it only supports binary masks, may fail to follow the text prompt (rows 1, 4, 5). Our models ("Ours") trained with Removal-Addition masks and training objectives generate edits with strong alignment to the text-prompts. Furthermore, with the masks, the models are able to make adjustments to traffic according to instruction (rows 1, 2, 3, 4, 6, 7).

Table 2: Models labeled "ours" are trained using all objectives on the real-world subset of the LangDriveEdit dataset, combined with unsupervised objectives on the NuScenes dataset. UltraEdit and CycleGAN-Turbo refer to pretrained models without any additional fine-tuning. We further compare UltraEdit-Text-SFT and UltraEdit-Mask-SFT, two variants trained with supervised fine-tuning but differing in how object changes are specified. In UltraEdit-Text-SFT, object positions are described exclusively through text, whereas in UltraEdit-Mask-SFT, object positions are conveyed using Removal-Addition masks. More details of their definition can be found in H.2. The best results for each setting are highlighted.

| Model | L1 ($\downarrow$) | L2 ($\downarrow$) | CLIP ($\uparrow$) | DINO ($\uparrow$) |
|---|---|---|---|---|
| **Bagel** | | | | |
| Bagel | 0.2245 | 0.0891 | 0.8399 | 0.7261 |
| **UltraEdit** | | | | |
| UltraEdit | 0.2282 | 0.0927 | 0.8475 | 0.7688 |
| UltraEdit-Text-SFT | 0.2336 | 0.1016 | 0.8583 | 0.7319 |
| UltraEdit-Mask-SFT | 0.1929 | 0.0676 | 0.8798 | 0.8173 |
| UltraEdit (Ours) | 0.1144 | 0.0296 | 0.9312 | 0.9024 |
| **CycleGAN-Turbo** | | | | |
| CycleGAN-Turbo | 0.1993 | 0.0649 | 0.8007 | 0.6378 |
| CycleGAN-Turbo (Ours) | 0.1401 | 0.0383 | 0.8800 | 0.8333 |

## 5.3 Fine-grained Instructions for Dense Driving Scenes

In Table 2, we also conduct an ablation study where we train a diffusion model that employ text with binary masks for localized edits ("UltraEdit-Text-SFT"). However, it can only perform on par with the base model. When we switch to using masks augmented with clip features to describe traffic patterns, we see an significant improvement in all scores suggesting higher instruction alignment in edited regions and content preservation in unedited regions. Our UltraEdit and our CycleGAN-Turbo each show a 40 % reduction in L1 and L2 compared to their second best counterparts. Adding unsupervised

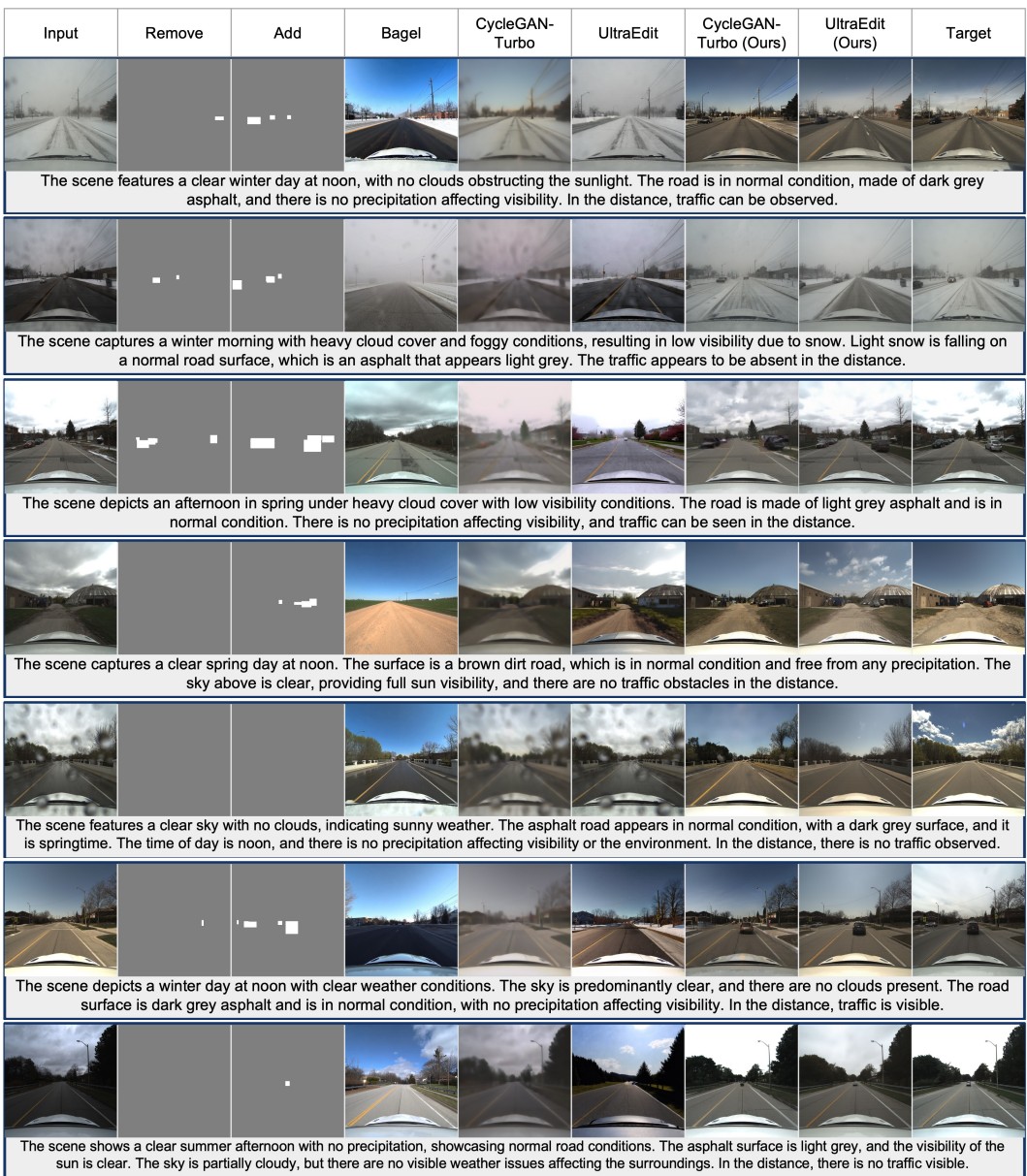

Figure 5: **Visualizations on Boreas**. Results of our models trained on Boreas compared to their baselines. The masks (projected as binary images) enable modifications to traffic while the text-prompt informs the desired global appearance.

training objectives to the Removal-Addition masks ("Ours") leads to a further improvement. An additional qualitative study of our loss functions in App. B.

## 5.4 EDITED STREET SCENES FOR DOWNSTREAM DRIVING TASKS

We apply our unsupervised losses to extend the method to NuScenes (Caesar et al., 2020), an out-of-distribution dataset. To this end, we jointly train an editing model on both the real-world Lang-DriveEdit dataset and NuScenes. We use this model to synthesize images from one random quarter of NuScenes under various weather conditions. Then we augment the original NuScenes quarter with these images and train two bird's-eye-view

Table 3: User Study. Overall preference distribution and win rates across 12 questions.

| Model | Pref. (%) | Win |
|---|---|---|
| CycleGAN-Turbo | 11.4 | 0.0 |
| Bagel | 15.9 | 8.3 |
| **Ours** | **72.7** | **91.7** |

| Baseline | Bagel | CycleGAN-Turbo | UltraEdit | CycleGAN-Turbo (Ours) | UltraEdit (Ours) |

The image presents a partially cloudy winter day with light cloud cover. It is noon, and the low visibility is due to sun coverage. The road surface is in normal condition.

The scene depicts a morning with normal road conditions and no rain impacting visibility. Overall, the weather is characterized as sunny and partly cloudy.

The area is enveloped by heavy cloud cover, with no rain affecting visibility. The road remains in normal condition.

Figure 6: **Out-of-Distribution Generations**. Side by side example of unedited and edited images by Bagel, UltraEdit, CycleGAN-Turbo, and our CycleGAN-Turbo and UltraEdit models jointly finetuned on our real-world dataset and NuScenes. We use the images produced by our model to train a bird-eye-view lane detection model.

(BEV) map segmentation models with and without synthesized images. Following (Liu et al., 2023), we report the highest IoU across different thresholds for each class separately in Table 4. This augmentation leads to an 33% improvement in the average of all classes compared to a baseline trained solely on NuScenes.

Table 4: BEV Map Segmentation. Intersection-over-Union (IoU) across 6 classes and the class-averaged IoU.

| Modality | Original | Augmented |
|---|---|---|
| Drivable Area | 0.5834 | 0.6448 |
| Ped. Crossing | 0.0533 | 0.1147 |
| Walkway | 0.1626 | 0.2251 |
| Stop Line | 0.0609 | 0.1059 |
| Carpark Area | 0.0981 | 0.1776 |
| Divider | 0.1569 | 0.2180 |
| **Mean** | **0.1859** | **0.2477** |

Figure 6 compares our synthetic NuScenes images with those from CycleGAN-Turbo, UltraEdit and Bagel. Our method shows the superior instruction alignment (rows 1, 2), whereas CycleGAN-Turbo produces blurry and misaligned outputs (rows 1, 3), UltraEdit doesn't edit the image (rows 1, 2, 3), and Bagel sometimes fails to maintain scene consistency (rows 1, 3).

We conducted a user study on the quality of our NuScenes edits produced by Bagel, CyclGAN-Turbo and Our trained CycleGAN-Turbo. We analyzed the responses of 17 participants over 12 images for a total of 176 responses. In each question, we asked participants to "select the better image based on the target scene description. Consider which image better matches the described target scene while maintaining image quality and realism." and "which image better satisfies the editing instruction while preserving the unedited parts of the scene?" As shown in Table 3 preferred our edits for balancing instruction alignment, content preservation and quality.

## 6 CONCLUSION

In this work, we present the LangDriveEdit dataset, a significant step forward in the simulation and evaluation of autonomous driving systems through language-driven image editing. By introducing a large-scale, paired dataset with fine-grained visual instructions and precise spatial masks, we enable more controllable, realistic, and diverse scene modifications. Experimental results across state-of-the-art models demonstrate marked improvements in both content preservation and instruction alignment, underscoring the dataset's utility. LangDriveEdit not only fills a critical gap in existing benchmarks but also opens new avenues for research in instruction-guided scene editing for safe and robust autonomous driving.

## THE USE OF LARGE LANGUAGE MODELS (LLMS)

LLMs did not play a significant role in either the research ideation or the writing of this paper. Their use was limited to correcting minor grammatical issues and typographical errors.

## LIMITATIONS

Our approach relies on language models for generating editing instructions, which may introduce hallucinations or factual inaccuracies. While our precise mask conditioning helps mitigate these issues by constraining modifications to specific regions, some inconsistencies remain. Future work could explore several robustness enhancements: implementing multi-step verification where an LLM validates initial instructions, incorporating human feedback through active learning or implementing automatic consistency checking between generated instructions and source images.

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

## A  RELATED WORKS

**Image Editing Dataset.**  Developing image-editing datasets is more challenging than Text-2-Image (Betker et al.; Ramesh et al., 2021; 2022; Rombach et al., 2022), with data scarcity being a major bottleneck (Wang et al., 2023a; Hui et al., 2024). MagicBrush (Zhang et al., 2024b) uses manual annotation with DALL-E2 (Ramesh et al., 2022), while InstructPix2Pix (Brooks et al., 2023a) generates pairs using the prompt-to-prompt method (Hertz et al., 2022) on LAION-Aesthetics (Schuhmann et al., 2022). SeedEdit (Shi et al., 2024) iteratively refines data and models. Most prior work targets simple object edits, underperforming in complex street scenes. We present *LangDriveEdit*, the first instruction-driven editing dataset for autonomous driving contexts.

**Image Editing via Generation.**  Instruction-based editing of real photos is a key task in image processing (Gao et al., 2024b; Crowson et al., 2022; Liu et al., 2020; Zhang et al., 2023a; Ruiz et al., 2023; Pan et al., 2024). Large-scale diffusion models have greatly enhanced text-driven editing (Kawar et al., 2022; Saharia et al.; Li et al., 2023; Chen et al., 2023; Ma et al., 2023; Meng et al., 2022; Mokady et al., 2023; Tumanyan et al., 2022; Nichol et al., 2022; Sheynin et al., 2023a). Recent models like InstructPix2Pix (Brooks et al., 2023a) and HIVE (Zhang et al., 2023b) allow users to edit images via instructions. MagicBrush (Zhang et al., 2024b) enhances this with manual annotations, and UltraEdit (Zhao et al., 2024) sets a new benchmark using synthetic data. We show our dataset further boosts these methods in street-scene editing.

**Image Editing for Autonomous Driving.** Rising demand for driving-scene data has led to scene editing methods using NeRF or Gaussian Splatting (Liang et al., 2025; Yang et al., 2023b; Sun et al., 2024; Tonderski et al., 2024; Chen et al., 2024b), though these struggle with diverse scene composition. Meanwhile, multi-condition generation methods are gaining interest (Swerdlow et al., 2024; Yang et al., 2023a; Wang et al., 2023b; Gao et al., 2023; Wen et al., 2024; Alhaija et al., 2025; Gao et al., 2024a; Lu et al., 2024). Yet, instruction-guided image editing for driving remains underexplored due to a lack of datasets. We introduce *LangDriveEdit* to fill this gap.

## B  ADDITIONAL RESULTS

We conduct an qualitative ablation study at Figure 7. Specifically, when training solely with SFT (column 2) the model cannot preserve the details of the small cars and merges them together (row 1). Meanwhile, when we add unsupervised training objectives with the exception of the CLIP loss (column 3), the model outputs degenerate to match the input (row 2) especially for Nuscene. Using SFT, with unsupervised objectives, and CLIP similarity loss to prevent degeneration (column 3) shows the highest degree of instruction following (row 2) and content preservation (rows 1, 2).

## C  SYNTHETIC DATASET DEVELOPMENT

### C.1  PRECISE CONTROL OF SCENE VARIATIONS IN SYNTHETIC ENVIRONMENTS

Our paired instructions and annotations on real images (Section 3.1) enable the learning of fine-grained edits across diverse objects and scenarios. Yet, real-world data remains limited in supporting arbitrary and precise manipulations. In contrast, synthetic environments allow flexible control over textures, semantics, and backgrounds, even for individual objects within a scene.

To provide complementary and precise control over scene variations, we additionally include paired images, annotations, and instructions from synthetic environments, as a supplement to our real-world images. We choose Carla (Dosovitskiy et al., 2017), an open-source simulator designed as a research and development platform for autonomous driving with extensive customization capabilities. With this simulator, we synthesize paired videos from the perspective of an autonomous vehicle across six diverse urban and suburban environments. Recent works demonstrate CARLA's utility in closed-loop end-to-end driving with language models (Shao et al., 2023), safety-critical scenario generation for autonomous vehicle testing (Zhang et al., 2024a), and multi-ability benchmarking of end-to-end driving systems (Jia et al., 2024).

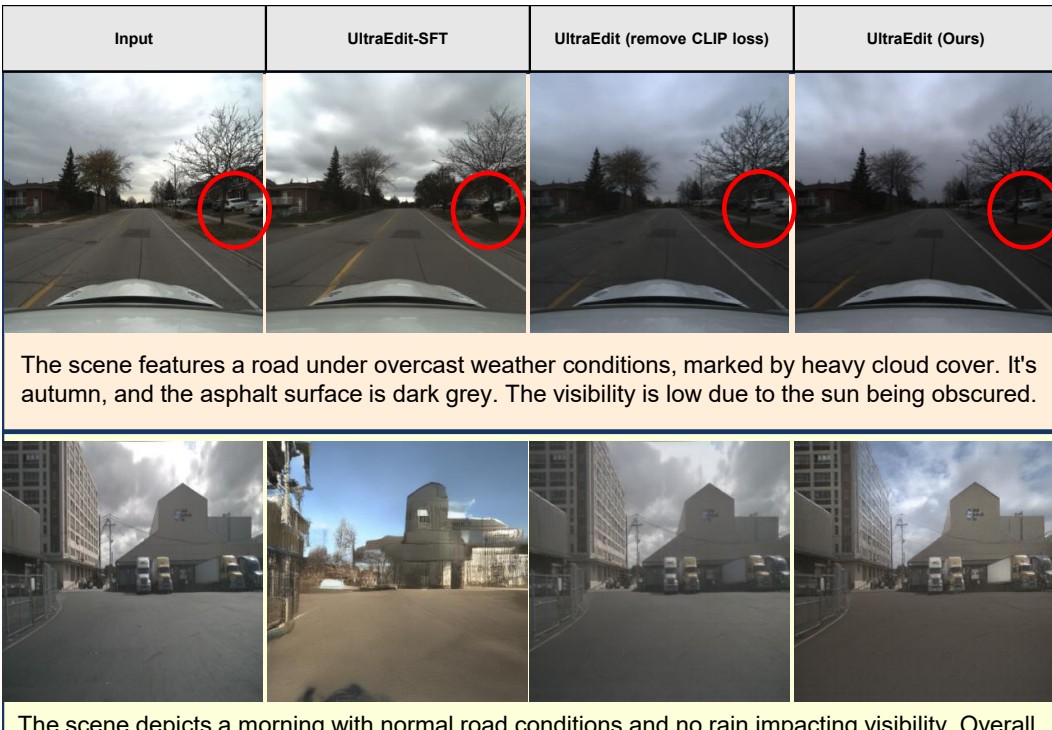

| Input | UltraEdit-SFT | UltraEdit (remove CLIP loss) | UltraEdit (Ours) |

The scene features a road under overcast weather conditions, marked by heavy cloud cover. It's autumn, and the asphalt surface is dark grey. The visibility is low due to the sun being obscured.

The scene depicts a morning with normal road conditions and no rain impacting visibility. Overall, the weather is characterized as sunny and partly cloudy.

Figure 7: **Ablation on Training Objectives**. Top: Boreas. Bottom: Nuscenes. We use blank masks during the evaluation of Boreas and study the content preservation abilities of the models over the vehicles.

Table 5: Comparison of edits performed by open-source driving scene editing datasets: Snow 100K (Liu et al., 2018), Outdoor-Rain (Out-Rain) (Li et al., 2019), Rain-Drop (Qian et al., 2018), MT Weather (Zhao et al., 2018), and *LangDriveEdit* (Ours) on Carla.

| Category | Edit | Snow 100K | Out-Rain | Rain-Drop | MT Weather | Ours (Carla) |
|---|---|:---:|:---:|:---:|:---:|:---:|
| Global | Time of Day | | | | | ✓ |
| | Weather | ✓ | ✓ | ✓ | ✓ | ✓ |
| | Season | ✓ | | | | ✓ |
| Road | Road Condition | | | | | ✓ |
| | Road Type | | | | | ✓ |
| Building | Building Appearance | | | | | ✓ |
| Vehicle | Vehicle Color | | | | | ✓ |
| | Vehicle Type | | | | | ✓ |
| | Vehicle Changes | | | | | ✓ |
| Traffic Signal | Traffic Light Color | | | | | ✓ |
| Pedestrian | Pedestrian Changes | | | | | ✓ |
| | Pedestrian Clothing | | | | | ✓ |

### C.1.1 Structured Scene Edits and Selections for Aligned Instruction

For each video pair, we implement controlled edits between the base and modified simulations, each spanning 30 seconds of driving. We list our edit types in Table 5. Specifically:

- The distribution and quantity of pedestrians and vehicles are randomly sampled to ensure diversity. Pedestrian edits are applied to all pedestrians in a scene, while the specific change (e.g., clothing style or color) is randomly sampled for each individual from the Carla pedestrian catalog. Text

descriptions of cataloged clothing were created through visual inspection. As of the time of this publication, we use the dev-branch of the Unreal Engine 4.26 version of CARLA and follow the pedestrian catalog at https://carla.readthedocs.io/en/latest/catalogue_pedestrians/.

- All vehicles receive the same type of modification, with random variations (e.g., color) applied individually.

- Weather conditions and time of day are selected from prepared Carla profiles to maintain realistic scenes.

- For environmental elements, we assign randomized textures to road surfaces and buildings, with each building receiving a distinct texture to enhance scene variability.

Each simulation captures comprehensive views through six ego-mounted cameras providing 360-degree coverage, all used in the dataset construction. The sensor suite records RGB images, depth maps, semantic segmentation maps, and instance segmentation maps for each frame. Following our workflow on real images (Section 3.1.2), we leverage these semantic maps to prepare precise masks of subjects targeted for editing; meanwhile, global environmental edits use blank masks since these changes apply to the entire scene. Additionally, we document per-frame attributes for all vehicles, pedetrians, and traffic signals visible to the ego vehicle, for precise tracking of object properties and behaviors.

### C.1.2 FILTERING AND BALANCING FRAME PAIRS FOR CONTENT PRESERVATION

To further improve the quality and diversity of edited images, we apply a multi-stage filtering process:

- Remove frames where edited subjects are too small or absent;

- Eliminate frames with high visual similarity based on SSIM thresholds and color histogram correlation;

- Discard redundant frames captured when the ego vehicle was stationary;

- For global modifications without object-specific edits (such as weather and time-of-day changes), we retain all non-redundant frames that pass our similarity thresholds based on SSIM and color histograms.

To ensure only relevant edits are kept, we filter out frames where the edited subjects are too small or absent. Specifically, a 2D bounding box must cover at least 0.7% of the image area for pedestrians, 1.2% for vehicles, and 0.3% for traffic lights. We also remove frames with high visual similarity to others, defined as SSIM > 0.97 or color histogram correlation > 0.5, to maintain scene diversity. We use a blank mask for global edits, such as changes in weather or time of day, since these modifications can affect all objects in the scene (e.g., through lighting changes). We show our synthetic dataset statistics in Table 6. To address class imbalance in our training data, we oversample underrepresented editing categories.

## D REAL DATASET DEVELOPMENT

We generate: (1) a *removal mask* for the source image, which identifies objects to be eliminated, and (2) an *addition mask* for the target image, which highlights regions designated for new object placement. Together, they support fine-grained object edits while preserving non-targeted regions

These pixel-level editing masks are systematically derived by comparing corresponding frame annotations via three rules:

- Distance-based filtering: Objects beyond 50 meters from the ego vehicle are excluded unless they occupy a significant image area.

- Truncation detection for undersized 2D bounding boxes near image boundaries.

- Occlusion handling: In complex traffic scenarios, overlapping vehicle bounding boxes are each preserved to maintain scene coherence.

Table 6: **Synthetic Dataset Statistics**. Breakdown of samples across object-level and global environment editing types.

| Edit Type | Count | Percentage |
|---|---|---|
| **Object Editing** (458,136 samples) | | |
| Road Texture | 153,654 | 7.69% |
| Building Texture | 3,662 | 0.18% |
| Walker Color | 28,726 | 1.44% |
| Walker Replacement | 39,166 | 1.96% |
| Walker Deletion | 34,460 | 1.72% |
| Vehicle Replacement | 45,806 | 2.29% |
| Traffic Light State | 24,050 | 1.20% |
| Vehicle Color | 64,300 | 3.22% |
| Vehicle Deletion | 64,312 | 3.22% |
| **Global Environment Editing** (1,539,370 samples) | | |
| Weather | 527,450 | 26.41% |
| Weather + Time of Day | 549,612 | 27.51% |
| Time of Day | 462,308 | 23.14% |
| **Total Samples** | **1,997,506** | 100% |

# E  ANNOTATION PIPELINE

## E.1  ANNOTATION PIPELINE PROMPT

The VLM is prompted with the instruction loaded from:

```
1  You are an expert in autonomous driving, specializing in analyzing
       traffic scenes. You receive a series of traffic images from the
       perspective of the ego car. Your task is to describe the driving
       environment, focusing on weather, lighting, road layout, and
       environment.
2
3  It is essential that you strictly follow the rules and instructions below
       . Any deviation from the specified structure or format will result in
        an invalid output.
4
5  STRICTLY follow Rules:
6   - You must strictly follow the dictionary structure provided above.
7   - Only use the specified terms for weather, light, road layout, and
        environment. Do not create your own terms.
8   - No additional information or categories should be added.
9   - You should strictly follow these instructions. If an object or element
         is not visible or does not exist in the scene, set the value to '
        None'. Ensure every field is filled with the appropriate value or '
        None'.
10  - STRICTLY ignore any text written on the image.
11
12
13  Output the result in the following dictionary format:
14
15  {
16    "surrounding_info": {
17      "weather": "[e.g., 'cloudy', 'sunny', 'rainy', 'fog', 'snowy']",
18      "road_layout": "[Choose from: 'straight road', 'curved road', '
           intersection', 'T-junction', 'ramp']",
19      "environment": "[Choose from: 'city street', 'country road', 'highway
          ', 'residential area']",
20      "sun_visibility_conditions": "[Choose from: 'clear', 'foggy', 'low
           visibility', 'hazy']",
21      "road_condition": "[Choose from: 'wet', 'icy', 'normal', 'debris', '
           potholes']",
```

```
22      "surface_type": "[Choose from: 'asphalt', 'gravel', 'dirt', 'concrete
            ']",
23      "surface_color": "[Choose from: 'light grey', 'dark grey', 'black', '
            brown']",
24      "time_of_the_day": "[Choose from: 'morning', 'midday', 'afternoon', '
            night', 'dawn', 'dusk'.]",
25      "precipitation_intensity": "[Choose from: 'none', 'light', 'moderate
            ', 'heavy', 'torrential'.]",
26      "precipitation_visibility_impact": "[Choose from: 'none', 'low', '
            moderate', 'high']",
27      "cloud_cover": "[Choose from: 'clear', 'light', 'moderate', 'heavy'.]
28      }
29  }
```

### E.2 EXAMPLE ANNOTATION

We show an annotated image and the output caption from the annotation pipeline

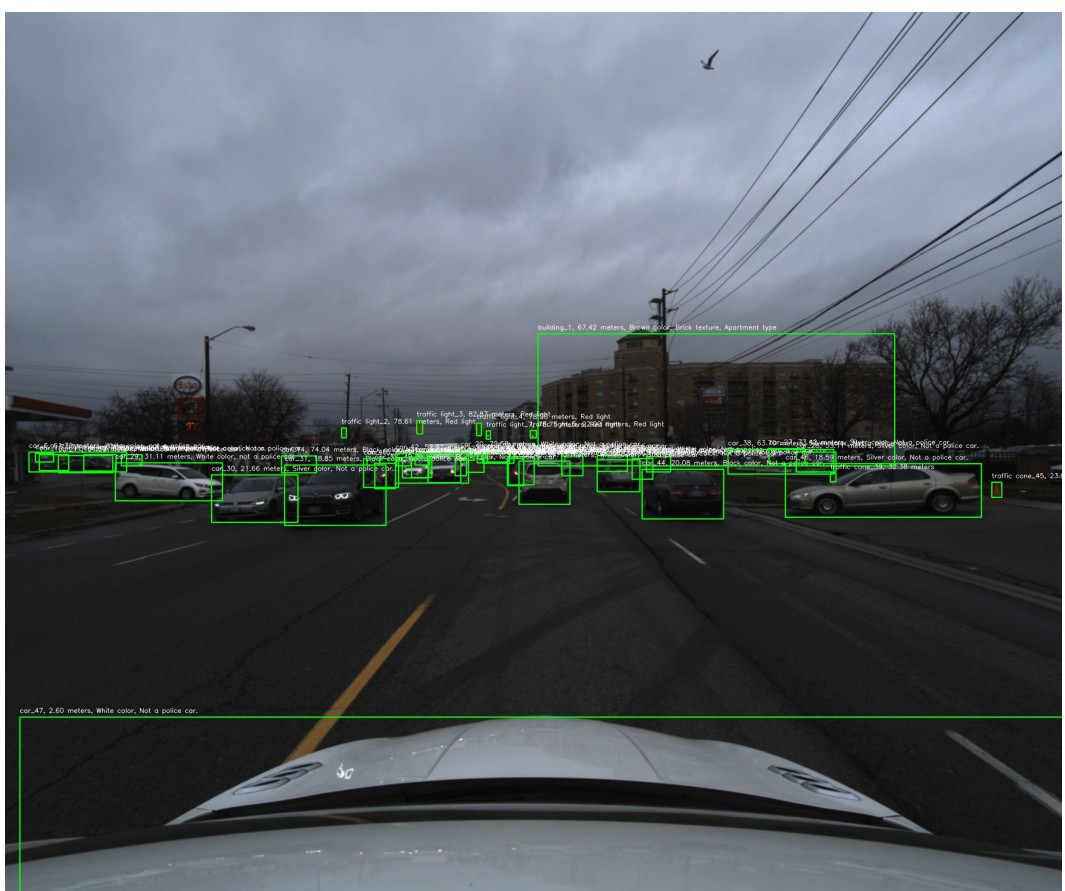

Figure 8: Example annotation of the image by the annotation pipeline.

We present a truncated version of the annotation for 8 below:

```
1  {
2      "surrounding_info": {
3          "weather": "cloudy",
4          "road_layout": "intersection",
5          "environment": "city street",
6          "sun_visibility_conditions": "low visibility",
7          "road_condition": "normal",
```

```
  8          "surface_type": "asphalt",
  9          "surface_color": "dark grey",
 10          "time_of_the_day": "morning",
 11          "precipitation_intensity": "none",
 12          "precipitation_visibility_impact": "none",
 13          "cloud_cover": "heavy"
 14      },
 15      "object_info": [
 16          {
 17              "class": "building",
 18              "bbox": [
 19                  1234,
 20                  745,
 21                  2060,
 22                  1051
 23              ],
 24              "object_id": 1,
 25              "distance_from_ego_vehicle": "67.42 meters",
 26              "attributes": "Brown color, Brick texture, Apartment type"
 27          },
 28          {
 29              "class": "traffic light",
 30              "bbox": [
 31                  779,
 32                  963,
 33                  790,
 34                  986
 35              ],
 36              "object_id": 2,
 37              "distance_from_ego_vehicle": "78.61 meters",
 38              "attributes": "Red light"
 39          },
 40          {
 41              "class": "car",
 42              "bbox": [
 43                  79,
 44                  1025,
 45                  147,
 46                  1062
 47              ],
 48              "object_id": 10,
 49              "distance_from_ego_vehicle": "69.81 meters",
 50              "attributes": "White color, Not a police car."
 51          },
 52          {
 53              "class": "car",
 54              "bbox": [
 55                  123,
 56                  1027,
 57                  249,
 58                  1067
 59              ],
 60              "object_id": 11,
 61              "distance_from_ego_vehicle": "61.59 meters",
 62              "attributes": "White color, Not a police car."
 63          }
 64      ]
 65 }
```

# F LLM GENERATED EDITING INSTRUCTIONS

## F.1 SYNTHETIC

We prompt chatGPT-4o mini with the following instruction to produce editing instructions for synthetic images based on the captions of the target image:

```
1  You are an expert in autonomous driving, specializing in analyzing
       traffic scenes. You receive a text description of a traffic image
       from the perspective of an autonomous vehicle's camera.
2
3  Your task is to produce FOUR VERSIONS of the SAME PROMPT, each with
       DIFFERENT WORDING BUT IDENTICAL CONTENT, that describes the driving
       scene depicted in the image.
4
5  IMPORTANT:
6  - Each version should describe the SAME SCENE, just phrased differently.
7  - Use natural, conversational language as if explaining the scene to
       another person.
8  - You may paraphrase, use synonyms, and vary sentence structure, but do
       not invent details not present in the caption.
9  - It is OK to combine or rephrase information for readability and flow.
10 - Avoid sounding like a computer or simply listing numbers and attributes
       . Make the description sound like something a human would say.
11 - Do not use quantitative descriptions. Only use qualitative natural
       language to describe the scene.
12 - Do NOT add your own subjective opinions or emotions (e.g., do not say '
       beautiful', 'moody', etc.), but you may use natural transitions and
       phrasing.
13
14
15 The output should be in the format below:
16
17 ### Scene Description:
18
19 version_1: {{description_1}}
20 version_2: {{description_2}}
21 version_3: {{description_3}}
22 version_4: {{description_4}}
23
24 Image Caption: {caption_0}
```

## F.2 REAL-WORLD

We prompt chatGPT-4o mini with the following instruction to produce editing instructions for real-world images based on the captions of the target image:

```
1  You are an expert in autonomous driving, specializing in analyzing
       traffic scenes. You receive a text description of a traffic image
       from the perspective of an autonomous vehicle's camera.
2
3  Your task is to produce FOUR VERSIONS of the SAME PROMPT, each with
       DIFFERENT WORDING BUT IDENTICAL CONTENT, that describes the driving
       scene depicted in the image.
4
5  IMPORTANT:
6  - Each version should contain the EXACT SAME DESCRIPTION, just phrased
       differently.
7  - ALL prompts should describe EXACTLY THE SAME SCENE with no variation in
        what is being described.
8  - Only use adjectives and descriptors that are explicitly provided in the
        caption. Do NOT add your own subjective descriptors like "moody," "
       tranquil," "charming," etc. Stick strictly to the attributes and
       descriptors that appear in the input caption.
```

```
 9
10  At the end of each prompt version, append the following line:
11      "There may be minor additional changes in time or weather (such as
            lighting, clouds, or rain) between the images that are not fully
            captured by the descriptions, but these are expected to be subtle
            ."
12
13  The prompt should be in the format below where each version describes the
        same contents but with different wording.
14
15  ### Scene Description:
16
17  version_1: {{description_1}}
18  version_2: {{description_2}}
19  version_3: {{description_3}}
20  version_4: {{description_4}}
21
22  Image Caption: {caption_0}
```

## G  MODEL AND EVALUATION DETAILS

We evaluate our training methods and pixel level instructions on two competitive image editing models: UltraEdit (Zhao et al., 2024)) and CycleGAN-Turbo (Parmar et al., 2024). CycleGAN-Turbo, based on the Stable Diffusion Turbo by Stability AI (Sauer et al., 2023) performs text-instructed image generation in one diffusion step. UltraEdit is based on Stable Diffusion 3 which is fine-tuned on 500K dataset of free-form edits and an additional 100K dataset of precise mask-conditioned edits. All models are evaluated with the dataset described in Sec. 3.

Following image editing benchmarks used in (Zhang et al., 2024c; Zhao et al., 2024; Sheynin et al., 2023b), we consider the following metrics: the L1 and L2 distance, the CLIP image similarity, and the DINO image similarity between the edited image and the ground truth. These metrics measure how well the edited image preserves the original content and reflects the required edit. For each of the real-world and synthetic datasets, we evaluate 2000 images for editing independently in both directions: transforming the source image to match the target, and conversely, modifying the target to reproduce the source.

## H  TRAINING DETAILS

### H.1  CYCLEGAN-TURBO

We train this model across $2 \times 80$GB NVIDIA A100 GPUs with a total batch size of 4 for 10000 steps. Our training parameters are:

$$
\begin{aligned}
\lambda_{\text{gan}} &= 0.5, \\
\lambda_{\text{id}} &= 0.05, \quad \lambda_{\text{id-lpips}} = 0.05, \\
\lambda_{\text{cycle}} &= 0.05, \quad \lambda_{\text{cycle-lpips}} = 0.05, \\
\lambda_{\text{sft}} &= 0.1, \quad \lambda_{\text{sft-lpips}} = 1.0, \\
\lambda_{\text{clip}} &= 0.5.
\end{aligned}
\tag{6}
$$

To incorporate the guidance from the remove and add masks, we expand the VAE input channels to accept the concatenation of the input image and conditioning masks and train end-to-end. The weights of any existing convolutions are maintained and new weights are initialized as zero. We train at a resolution of $512 \times 512$ and a learning rate of $1 \times 10^{-5}$.

### H.2  ULTRAEDIT

We evaluate the UltraEdit model with the following settings:

- **UltraEdit**. The UltraEdit model supports a single binary mask as conditioning therefore we simplify our remove and add masks into one with the union of their binary projections.

- **UltraEdit-text**. We train an UltraEdit model using only supervised objectives on a modified real-world subset of the LangDriveEdit dataset without CLIP masks. To do this, following the process described in Sec. 3.1.2, we construct the editing prompts by asking chatGPT-4o to describe, in addition to global changes, all objects to remove from left to right, and all objects to add from left to right. The pixel add and remove masks are the binary image equivalents of the full add and remove masks. This model is trained across $4 \times 48$ GB NVIDIA A6000 GPUs with a total batch size of 256 for 10000 steps.

- **UltraEdit-clip**. We train an UltraEdit model using only supervised training objectives and the real-world dataset described in Sec. 3. We train this model across $4 \times 80$GB NVIDIA A100 GPUs with a total batch size of 4 for 5000 steps.

- **UltraEdit (ours)** We train an UltraEdit model using the objectives described in Sec. 4 on our real-world dataset described in Sec. 3 in addition to unsupervised objectives on NuScenes. To adapt these objectives to multi-step diffusion, we apply gradient checkpointing and perform end-to-end training with our unsupervised losses. We train this model across $4 \times 80$GB NVIDIA A100 GPUs with a total batch size of 4 for 5000 steps. Our training parameters are:

$$
\begin{aligned}
\lambda_{\text{gan}} &= 0.5, \\
\lambda_{\text{id}} &= 0.05, & \lambda_{\text{id-lpips}} &= 0.05, \\
\lambda_{\text{cycle}} &= 0.05, & \lambda_{\text{cycle-lpips}} &= 0.05, \\
\lambda_{\text{sft}} &= 3.0, & \lambda_{\text{sft-lpips}} &= 0.5, \\
\lambda_{\text{clip}} &= 0.5.
\end{aligned}
\tag{7}
$$

### H.3 ROAD SEGMENTATION MODEL

We train the base model on a random quarter of the NuScenes dataset across $4 \times 48$GB NVIDIA A6000 GPUs with a total batch size of 16 for 10000 steps. We train for 20 epochs.

We train another model on our synthetic NuScenes dataset across $4 \times 48$GB NVIDIA A6000 GPUs with a total batch size of 16 for 10000 steps for 20 epochs. Then we finetune on the original quarter of the NuScenes dataset across $4 \times 48$GB NVIDIA A6000 GPUs with a total batch size of 16 for 10000 steps for 20 epochs.

## I GENERATION DETAILS

### I.1 CYCLEGAN-TURBO

We maintain the default parameters from (Parmar et al., 2024) and evaluate our trained model with remove and add masks. The base model is evaluated without mask with the default parameters from (Parmar et al., 2024).

### I.1.1 ULTRAEDIT

- **UltraEdit**. The UltraEdit model supports a single binary mask as conditioning therefore we simplify our remove and add masks into one with the union of their binary projections. We maintain the default parameters from (Zhao et al., 2024).

- **UltraEdit-text**. We train an UltraEdit model using only supervised objectives on a modified real-world subset of the LangDriveEdit dataset with masks projected to binary images. We maintain the default parameters from (Zhao et al., 2024).

- **UltraEdit-clip**. We evaluate the model with full add and remove masks using 20 diffusion steps and classifier-free guidance scale of 1 and image guidance scale of 1.

- **UltraEdit (ours)** We evaluate the model with full add and remove masks using 8 diffusion steps and classifier-free guidance scale of 1 and image guidance scale of 1.

