# OpenReview forum: "LangDriveEdit: Language-Driven Image Editing for Street Scenes"
_ICLR.cc/2026/Conference — ICLR 2026 Conference Withdrawn Submission_

### Official Review · Reviewer_Ws7H · 2025-10-26

**Soundness:** 3
**Presentation:** 2
**Contribution:** 3
**Rating:** 4
**Confidence:** 4

**Summary:**

This paper introduces LangDriveEdit, a new dataset for language-driven image editing tasks, specifically designed for autonomous driving scenes. Authors provide details about how the dataset is curated from Boreas, and can be used to finetuning image-editing tasks for better content preservation and instruction alignment. They also show that edited images produced with our dataset can improve road segmentation performance on an out-of-distribution driving dataset.

**Strengths:**

1. This dataset is the first instruction-driven editing dataset tailored to autonomous driving scenarios, which is useful for the autonomous driving community. The annotation is rich and dense, including global text descriptions and fine-grained object masks with clip features.
2. Propose a training strategy for finetuning image-editing models with LangDriveEdit dataset for better content preservation and instruction alignment. Experiments shows that it significantly makes models like UltraEdit and CycleGAN-Turbo better in driving-specific editing task.

**Weaknesses:**

1. The whole dataset curation pipeline heavily relies on pretrained models, LLMs and VLMs, and no human annotation is used. The wrong inference results from pretrained models can directly harm the quality of the dataset. Authors should analyze the failure cases and how to ensure the quality of annotations.
2. Seems like $t_s$ and $t_t$ are independent captions for source image and target image, not a real transformation caption like $t_{s\rightarrow t}$.
3. Since are no instructions for the removal and appending objects in the text prompt, the capability of object editing strongly depends on the masks, which limits the usability of the dataset. This is fatal, we cannot expect the user to provide such masks.

**Questions:**

1. The usage of `Expanding Masks with CLIP Text Features` is not clear. Authors provide such annotations but not use them later.

---

### Official Review · Reviewer_2tE2 · 2025-10-30

**Soundness:** 3
**Presentation:** 3
**Contribution:** 3
**Rating:** 6
**Confidence:** 4

**Summary:**

The paper introduces LangDriveEdit, the first dataset designed specifically for language-driven editing of autonomous driving scenes, aiming to enhance content preservation and instruction alignment in generative models. The dataset combines real-world and synthetic images, each paired with fine-grained textual and visual editing instructions across 12 types of global and local transformations. Experimental results show substantial gains in prompt alignment, visual fidelity, and content preservation when fine-tuning state-of-the-art diffusion models on LangDriveEdit.

**Strengths:**

1. The paper presents LangDriveEdit, the first dedicated dataset for text-guided editing of driving scenes, filling a major gap in generative data resources for autonomous driving research.

2. The integration of vision–language models, depth estimation, and mask generation creates a scalable, semi-automatic pipeline for producing paired visual and textual instructions, which could be reused by future work.

3. By maintaining semantic fidelity in edited driving scenes, the dataset enables realistic simulation of diverse environmental conditions, supporting safer and more robust model evaluation within an Operational Design Domain.

**Weaknesses:**

1. In Section 4.2, it remains unclear how the language instructions describing transformations (from source to target and back) are obtained automatically. The paper should explain how it ensures sufficient diversity to prevent the model from exploiting statistical shortcuts during self-supervision.

2. The framework focuses exclusively on static images, while real autonomous driving systems depend on video streams. The lack of temporal modeling limits its applicability to real-world driving scenarios that require temporal and multi-view consistency.

3. Table 2 only evaluates three models, with the proposed fine-tuning applied to just two, making the comparative analysis statistically weak. Including more baselines and reporting additional metrics (e.g., IoU for foreground objects) would give a clearer picture of the model’s editing precision.

4. The paper lacks comparisons with recent diffusion-based scene editing approaches and omits standard perceptual quality metrics such as FID, which would be important for quantifying image realism.

5. Missing reference: SimGen (NeurIPS 2024).

**Questions:**

Please refer to the weaknesses above.

---

### Official Review · Reviewer_mpx4 · 2025-10-30

**Soundness:** 2
**Presentation:** 3
**Contribution:** 2
**Rating:** 4
**Confidence:** 5

**Summary:**

The paper proposes LangDriveEdit, a dataset and training pipeline for instruction-guided image editing in autonomous-driving street scenes. The main claim is that existing instruction-based editing models (UltraEdit, etc.) don’t enforce two properties that are crucial for driving data: (i) content preservation (do not destroy lane lines, traffic lights, vehicles that should stay) and (ii) instruction alignment (actually do the weather/time/traffic/object edits requested by the text). The authors build a paired dataset by aligning multi-season Boreas scenes (same route, different conditions) and generating LLM-based instructions and pixel-level remove/add masks via a multi-modal annotation pipeline (VLM + depth + detection + SAM). They also add a synthetic CARLA part with 12 controllable edit types (weather/time/vehicle/pedestrian/road), also paired with instructions and masks. Then they propose to fine-tune existing editing models (UltraEdit, CycleGAN-Turbo) with a combo of three losses: supervised fine-tuning (SFT) for the paired cases, language-guided cycle for content preservation on unpaired data, and language-guided CLIP loss to avoid degenerating to identity. On their dataset, their fine-tuned models get lower L1/L2 and higher CLIP/DINO than off-the-shelf baselines, and they also show a small BEV map segmentation gain on an augmented nuScenes split.

**Strengths:**

* The combination of SFT (when you have source–target) and language-guided cycle + identity (when you don’t) is a nice way.

**Weaknesses:**

* Right now the driving/simulation community is moving to 4D / video / multi-view / scene-level editing and generation. This task stays at single-frame image editing (even though the data is collected from sequences), and there is no viewpoint-consistent or temporally-consistent editing. That makes the story weaker for “safety” and for “ODD expansion”, because real validation pipelines need temporally coherent, multi-camera data.
* A single BEV segmentation experiment on nuScenes is not enough to claim “edited driving scenes improve safety-critical downstream tasks.” Modern AD sim / generative papers usually at least try: det → HD map → tracking → planning / closed loop on edited or generated data, or they plug into an existing stack to show influence on perception → prediction → planning.
* The teaser (Fig. 1) and the comparisons (Fig. 5, Fig. 6) don’t look good in terms of visual pleasantness and faithful instruction following.
* The viewpoints are actually weakly aligned.
* The CARLA part is huge (∼2M samples), but CARLA’s texture, lighting, and object models are obviously synthetic compared to Boreas / nuScenes.
* Bagel is used off-the-shelf while the authors fine-tune UltraEdit / CycleGAN-Turbo on their own data + losses.

**Questions:**

See weakness

---

### Official Review · Reviewer_khDT · 2025-11-01

**Soundness:** 2
**Presentation:** 3
**Contribution:** 2
**Rating:** 4
**Confidence:** 4

**Summary:**

This article introduces a new dataset called LangDriveEdit, designed specifically for language driven image editing of autonomous driving street scenes. This dataset combines real-world data (from Boreas) and synthetic data (from CARLA), supporting 12 editing tasks. The paper aims to address two core issues: 1) content preservation (e.g. not changing lane markings when editing weather) and 2) instruction alignment (e.g. faithfully executing complex edits). To address this issue, their dataset provides fine-grained instructions, including not only text prompts but also pixel level visual instructions in the form of "remove" and "add" masks. This paper proposes a set of training objectives to fine tune existing editing models.

**Strengths:**

The paper is well written.

The paper addresses a key challenge in autonomous driving — paired driving data for editing driving scenes.

The paper shows a 33% relative mIoU improvement in BEV segmentation when using generated data.

**Weaknesses:**

The claim of being the first dataset for language-driven scene editing is weak, as prior works (e.g., SceneCrafter) have similar data.

The “paired” images are just matched by camera pose at different times. Random traffic differences are treated as edits, which are not semantically meaningful. Edits are only based on 2D masks without 3D information, limiting use for 3D-aware tasks like planning.

Focusing on single images ignores video, which is crucial for temporal consistency in driving.

The annotation pipeline depends on multiple models (e.g., Owlv2, Metric3D, SAM) but no accuracy metrics are provided, so label quality is uncertain.

Missing comparisons with key baselines (e.g., InstructPix2Pix, Stable Diffusion Inpainting). Current results only show that fine-tuning helps, not that the method is superior.

The training losses used are all standard (L1/LPIPS, cycle, CLIP). The method feels incremental rather than truly novel.

**Questions:**

Regarding Figure 3 (left), are the metric coordinates (e.g., "42.09 meters") actually part of the text promptfed to the model? If so, how does the model learn to ground these precise, continuous metric values?

Please quantify the error rate of the "Image Descriptor" pipeline. Specifically, what is the mAP for object detection, the mean error for depth, and the attribute classification accuracy?

Will the dataset and theannotation pipeline be made publicly available?

---

### Note · Authors · 2025-11-15

I have read and agree with the venue's withdrawal policy on behalf of myself and my co-authors.